# Preparation and Characterization of Moldable Demineralized Bone Matrix/Calcium Sulfate Composite Bone Graft Materials

**DOI:** 10.3390/jfb12040056

**Published:** 2021-10-04

**Authors:** I-Cheng Chen, Chen-Ying Su, Chun-Cheih Lai, Yi-Syue Tsou, Yudong Zheng, Hsu-Wei Fang

**Affiliations:** 1Accelerator for Happiness and Health Industry, National Taipei University of Technology, Taipei 10608, Taiwan; icchen.ntut@mail.ntut.edu.tw; 2Department of Chemical Engineering and Biotechnology, National Taipei University of Technology, Taipei 10608, Taiwan; chenying.su@mail.ntut.edu.tw (C.-Y.S.); arashimaple@yahoo.com.tw (C.-C.L.); 3Department of Neurosurgery, Taipei Medical University Hospital, Taipei 11031, Taiwan; b101091102@tmu.edu.tw; 4Taipei Neuroscience Institute, Taipei Medical University, Taipei 11031, Taiwan; 5Neural Regenerative Medicine, College of Medical Science and Technology, Taipei Medical University and National Health Research Institute, Taipei 11031, Taiwan; 6School of Materials Science and Engineering, University of Science and Technology Beijing, Beijing 100083, China; 7Institute of Biomedical Engineering and Nanomedicine, National Health Research Institutes, Miaoli County 35053, Taiwan

**Keywords:** DBM, demineralized bone matrix, calcium sulfate, putty, moldable, bioceramics, biocomposites, bone implants

## Abstract

Demineralized bone matrix (DBM) is a decalcified allo/xenograft retaining collagen and noncollagenous proteins, which has been extensively used because of its osteoconductive and osteoinductive properties. Calcium sulfate (CaSO_4_, CS) is a synthetic bone substitute used in bone healing with biocompatible, nontoxic, bioabsorbable, osteoconductive, and good mechanical characteristics. This study aims to prepare a DBM/CS composite bone graft material in a moldable putty form without compromising the peculiar properties of DBM and CS. For this purpose, firstly, porcine femur was defatted using chloroform/methanol and extracted by acid for demineralization, then freeze-dried and milled/sieved to obtain DBM powder. Secondly, the α-form and β-form of calcium sulfate hemihydrate (CaSO_4_·0.5H_2_O, CSH) were produced by heating gypsum (CaSO_4_·2H_2_O). The morphology and particle sizes of α- and β-CSH were obtained by SEM, and their chemical properties were confirmed by EDS, FTIR and XRD. Furthermore, the DBM-based graft was mixed with α- or β-CSH at a ratio of 9:1, and glycerol/4% HPMC was added as a carrier to produce a putty. DBM/CSH putty possesses a low washout rate, good mechanical strength and biocompatibility. In conclusion, we believe that the moldable DBM/CSH composite putty developed in this study could be a promising substitute for the currently available bone grafts, and might have practical application in the orthopedics field as a potential bone void filler.

## 1. Introduction

Bone defects can occur from various causes such as aging, tumor, trauma, infection, surgery, or other congenital diseases [1]. Bone grafting is one of the most commonly used strategies for treating bone defects, in which bone grafts are widely applied for bone regeneration in orthopaedic surgeries [2]. An ideal bone graft must augment the process of bone healing with optimal stability and durability, along with characteristics including osteogenesis, osteoinduction, and osteoconduction [3]. Currently, autologous bone graft is the only material that fulfills the aforementioned criteria, and represents the “gold standard” clinical graft for bone defect repair with no evidence of graft rejection or histocompatibility issues. However, the shortage of supply, donor site morbidity, extended hospital length of stay and, most notably, the increased risk of surgical site infections has limited widespread application in the clinical practice. This has prompted researchers to explore potential alternatives such as synthetic, allograft and xenograft bone substitutes. These bone substitutes have certain advantages such as no supply limitation due to more availability of donors (in the case of allo and xenografts), whereas the synthetic bone substitutes can be manufactured with desired quantity and quality [4,5].

Allograft or allogeneic transplant is medical terminology referring to the transfer of bone harvested using bone tissue collected from a genetically nonidentical donor to a recipient of the same species, whereas xenograft refers to the transfer of bone tissue between different species. These are considered to be alternatives for autografts due to the similarity in chemical components and physical structure, which manifest good osteoinductive and osteoconductive properties. However, the risks of immune responses and disease transmission remains the major drawback. Demineralized bone matrix (DBM) is a kind of acid extracted demineralized allograft derivative composed of collagens, noncollagenous proteins and growth factors [6]. After decalcification, growth factors can be released from the surrounding mineral components and fully exert their osteoinductive potential, and the remaining collagen provides a 3D configuration for osteoconduction [7]. Based on different manufacturing techniques, DBM is available in different types such as sponges, freeze-dried powder, gel, paste, putty, or strips. Among them, DBM powder possesses a large surface area for exposing collagen or other growth factors at the graft site, exerting superb osteoinductive ability [8,9]. However, lack of mechanical strength and stability, as well as difficulties in handling, are the major drawbacks of DBM powder for clinical use [10,11]. 

Calcium sulfate (CS) is a kind of ceramic-based synthetic bone graft material with a long history of use for bone healing since the 19th century [12]. Gypsum mainly consists of calcium sulfate dihydrate (CSD, CaSO_4_·2H_2_O), which can be a promising raw material to produce calcium sulfate hemihydrate (CSH, CaSO_4_·0.5H_2_O, Plaster of Paris). When gypsum is heated to over 110 °C, water can be removed in a process known as calcination, resulting in CSH production. CSH exists in α and β forms with identical chemical properties but presenting great variation in terms of structure, crystal size, surface area, and lattice imperfections [13]. CSH is known to be suitable for clinical applications, such as a bone void filler, since it possesses key features including osteoconductivity, excellent workability, high self-setting strength, biocompatibility, rapid and complete resorption with minimal inflammation, and is relatively cost effective. Moreover, extensive studies have shown that CSH also exerts an osteoinductive property, in which, the calcium ions may be released during the dissolution of CS, which affects osteoblast genesis and function [13,14,15,16,17]. The most cited limitations towards the clinical application of CS bone grafts are their rapid resorptive rates compared to than the new bone growth, which is due to their higher solubility in physiological body fluids, resulting in delayed bone healing [16,18].

CSH is very suitable for filling and repairing bone defects due to its plasticity and self-hardening properties. Although the rapid biodegradation rate limits its clinical application, some studies have provided approaches by which CS bone grafting systems can be modified with other materials as composites to be more durable with enhanced bone regenerative potential [19,20]. For example, some studies have reported composite bone grafting systems combining CS possessing osteogenic potential with the slower resorption times of synthetic hydroxyapatite grafts [21,22]. Furthermore, composite bone graft materials combined with biocompatible viscous carriers, including natural and synthetic resources, could produce moldable products, such as paste or putty, to facilitate handling and packing of these materials into defect sites [8]. 

Since they are imported from other countries, most synthetic bone graft substitutes are expensive in Taiwan; therefore, in the present study, we intended to develop a local moldable composite bone graft product which is cost-effective, easy to handle, readily available and with good biocompatibility. As DBM possess excellent osteoinductive and osteoconductive capability, and CSH provides mechanical strength with self-hardening properties, a composite DBM/CSH putty was prepared and the physical/chemical properties, setting time, mechanical strength, in vitro bioactivity, and biocompatibility were evaluated. The procedures and results provided from this study could be used potentially for the production of the very first domestic DBM/CSH putty as a bone void filler for effective bone healing. 

## 2. Materials and Methods

### 2.1. DBM Powder Preparation

The procedure for preparation of a DBM implant is shown in Figure 1. The cortical bone was harvested from the femurs of a porcine species, and soft tissues were removed. The bones were cut into small pieces and then washed in distilled water to remove remaining bone marrow and blood. The bone segments were dehydrated in ethanol for 2 h and the lipid was removed by a chloroform/methanol (1:1) reagent or diethyl ester for 12 h. Defatted bones were washed in distilled water and demineralization was accomplished by adding 0.6 N HCl or acidic AlCl_3_ reagent (containing 0.5 M AlCl_3_, 3% HCl, and 5% formic acid) for 5 days (different treating protocols were shown in Table 1). The demineralized specimen was neutralized by 5% Na_2_SO_4_ then washed in distilled water for overnight. The bone material was dehydrated in ethanol again for 2 h then pulverized into smaller pieces. Samples were decellularized by treating samples with 0.05% trypsin-EDTA solution for 24 h. After freeze-drying for 48 h, the specimen was milled by a high-efficiency ball mill (90s; 8000 M Mixer, SPEX, Metuchen, NJ, USA), planetary ball mill (350 rpm, 5 min; PM 100, Retsch, Haan, Germany), blade or diamond file. Chloroform, methanol, diethyl ester, HCl, AlCl_3_, Na_2_SO_4_, trypsin-EDTA solution were purchased from Sigma (Sigma, St. Louis, MO, USA), and formic acid was obtained from J. T. Baker (Avantor, Inc., Radnor, PA, USA).

### 2.2. dsDNA Measurement

Lyophilized DBM powder (10 mg) was treated with 1 mg/mL of pepsin (Sigma, St. Louis, MO, USA) in 0.01 N HCl for 96 h at room temperature until no visible matrix remained, then centrifuged at 14000 rpm for 5 min to collect supernatant for dsDNA Quantitation. ds DNA was measured by a Quant-iT™ PicoGreen™ dsDNA Assay Kit (Thermo Fisher Scientific, Waltham, MA, USA) following the manufacturer’s instruction. The samples were read by Thermo Scientific Varioskan Flash (Thermo Fisher Scientific, Waltham, MA, USA) to obtain OD values (excitation: 480 nm, emission: 520 nm).

### 2.3. Collagen Content Measurement

Lyophilized DBM powder (10 mg) was treated with 1 mg/mL of pepsin in 0.01 N HCl for 96 h at room temperature until no visible matrix remained, then centrifuged at 14,000 rpm for 5 min to collect supernatant for collagen quantification. The sample was acid hydrolyzed with equal volume of 37% HCl at 110 °C overnight and treated with NaOH. Fifty μL of the resultant sample was mixed with 450 μL chloramine T reagent (0.07 M Chloramine-T in 10% *v/v* 1-propanol, 0.7 M NaOH, 0.2 M citric acid, 0.425 M sodium acetate trihydrate, and 1% glacial acetic acid) for 25 min. Five-hundred μL of Ehrlich’s solution (1 M 4-(Dimethylamino)benzaldehyde in 33% perchronic acid and 67% 1-propanol) was then added to the sample and incubated at 65 °C for 20 min. Standards (hydroxyproline) and samples were plated in triplicate in a 96-well clear, flat-bottomed plate (200 μL/well) and read in a spectrophotometer Thermo Scientific Varioskan Flash (Thermo Fisher Scientific, Waltham, MA, USA) at an absorbance wavelength of 550 nm. Chloramine-T, 1-propanol, glacial acetic acid, 4-(Dimethylamino)benzaldehyde, perchronic acid were purchased from Sigma (St. Louis, MO, USA), NaOH was from Showa (Tokyo, Japan) and citric acid, sodium acetate trihydrate were obtained from J.T Baker (Avantor, Inc., Radnor, PA, USA).

### 2.4. Calcium Sulfate Hemihydrate Preparation

The α-form of CSH (CaSO_4_·0.5H_2_O) was prepared using a wet, hot method. CSD (CaSO_4_·2H_2_O; J.T. Baker, Phillipsburg, NJ, USA) was heated at 132 °C for 30 min, then mixed with 30% CaCl_2_ (SHOWA Corporation, Osaka, Japan) solution and heated at 132 °C for 30 min again. Afterwards, boiling water was added to remove CaCl_2,_ and the mixture was incubated at 132 °C for 30 min. The resultant product was dried in oven at 50 °C then milled by a high-efficiency ball mill for 4 min [23,24] (Figure 2).

The β-form of CSH was prepared using a dry, hot method. CSD was heated at 132 °C for 30 min, autoclaved for 30 min then dried at 50 °C overnight. The β-form of CSH was obtained by milling with a high-efficiency ball mill for 4 min (Figure 2).

### 2.5. DBM/CSH Putty Preparation

DBM powder was mixed with CSH powder in the ratios of 1:1, 7:3, 8:2 or 9:1. The mixed DBM/CSH powder was then loaded in a carrier of 70% glycerol (Showa Chemical Industry Co., Ltd., Tokyo, Japan) with 0–4% HPMC (Sigma, St. Louis, MO, USA). The self-hardening, working time and setting time properties of the putty were evaluated (Figure 3). 

### 2.6. Scanning Electron Microscope (SEM) and Energy Dispersive Spectrometer (EDS)

Representative images and particle sizes of DBM powder and CSH powder were obtained by scanning electronic microscopy (SEM) using an S-3000H microscope (Hitachi, Tokyo, Japan) under low vacuum conditions. Each specimen was covered with gold by a sputter coater (Ion Sputter E101, Hitachi). An Energy Dispersive Spectrometer (EDS) X-act, Oxford Instruments, High Wycombe, UK) was used to detect the elements from preparation of DBM powder and synthesis of CSH.

### 2.7. Infrared Spectroscopy (FTIR)

The IR spectra of the samples prepared were recorded using a Nicolet AVATAR 330 Fourier Transform Infrared spectrophotometer (Thermo Electron, Waltham, MA, USA) with KBr mixed with sample at a 100:1 ratio. 

### 2.8. X-ray Diffraction (XRD)

X-ray diffraction (DMX-220, Rigaku, Tokyo, Japan) was performed to investigate the crystalline phases of synthetic CSH using Cu Kα X-rays generated at 40 kV and 30 mA at a diffraction angle (2θ) from 10° to 60° with a step size of 0.05°/step and an interval of 0.2 s/step.

### 2.9. Mechanical Compression Testing

To prepare the samples for the compressive strength test, the powder part (DBM, CSH and HPMC) and liquid part (glycerol and water) were mixed at a ratio of 0.25 (mL/g) under atmospheric conditions. Upon reaching the dough phase, the putty was spatulated into stainless steel molds and the resulting cylindrical samples had a diameter of 6 mm and a height of 12 mm for the mechanical test. The compressive strength of DBM/CSH composite putty was examined in accordance with ASTM F1633 standards. The compressive properties of cylindrical samples were measured at a loading rate of 1 mm/min with a universal mechanical testing machine (Transcell Technology Inc., New Taipei City, Taiwan). The measurements were performed three times for each group.

### 2.10. Washout Resistant Test

DBM/CSH putty was soaked in phosphate-buffer saline (PBS pH 7.4, Gibco, Thermo Fisher Scientific Inc., Waltham, MA, USA) for 15 min and statically placed in an incubator at 37 °C for 3 days. The weight loss ratio (washout) was calculated from the formula of the weight loss of the putty divided by the weight of initial sample. The reference putty used for the wash out resistant test was Genex® putty (Biocomposites Ltd., Staffordshire, UK).

### 2.11. Cell Viability

L929 cells (mouse fibroblasts, strain number BCRC 60091) were purchased from the Food Industry Research and Development Institute, Hsinchu, Taiwan. Cells were routinely maintained in Modified Eagle Medium (MEM, Gibco, Thermo Fisher Scientific Inc., Waltham, MA, USA) supplemented with 10% fetal bovine serum (FBS, Invitrogen, Waltham, MA, USA) at 37 °C under 5% CO_2_ and 95% relative humidity. 

In vitro cell viability testing of bone cement samples was conducted according to ISO10993-5. Briefly, DMB/DSH putty was extracted in 5 volumes of MEM at 37 °C for 24 h. L929 cells were plated at a density of 5 × 10^3^ cells/well onto a 96-well culture plate in culture medium at 37°C overnight. After removing culture medium the next day, cells were washed with PBS and cultured in MEM supplemented with 1% FBS containing DBM/putty extract. 3-(4,5-dimethylthiazol-2-yl)-2,5-diphenyltetrazolium bromide (MTT, Sigma, St. Louis, MO, USA) solution was then added to the medium, and cells were incubated at 37 °C for 2 h. The samples were read by an Enzyme-linked immunosorbent assay (ELISA) reader (Tecan, Männedorf, Switzerland) with a wavelength of 570 nm to obtain OD values. Cell viability higher than 70% was considered good biocompatibility.

## 3. Results

### 3.1. DBM Preparation

#### 3.1.1. Preparation of DBM Powder

The procedure for preparing DBM was dehydrating, defatting, demineralization, freeze-drying and milling/sieving (Figure 1 and Figure 4). The porcine femur was first chopped, washed and dehydrated by 99% ethanol. Selection agents for the preparation of DBM is critical. To obtain the optimal reagents for DBM powder preparation, two defatting reagents namely, chloroform/methanol (Chl/Met) or diethyl ester (Et_2_O), and two demineralization reagents, namely acidic AlCl_3_ reagent or 0.6 N HCl, were examined in this study (Table 1). Removal of fat was accomplished by incubating the small bone fragments in organic solvents such as chloroform/methanol or diethyl ester solution for 12 h. At this stage, chloroform/methanol was more effective in removing lipid because sample treated with diethyl ester still contained lipid residue. Next, demineralization was performed by treating samples with acid for 5 days. Strong acids such as HCl are often used decalcification agents. The end-point of demineralization was determined by inserting a scalpel directly into the specimen. Samples treated with 0.6 N HCl showed incomplete demineralization at day 5 were left longer, until day 20, for complete demineralization. By contrast, a faster rate of decalcification was observed in the samples treated with acidic AlCl_3_ reagent, which was completed within 5 days.

To obtain homogeneous DBM powder with the desired size range, different methods were tested for grinding (Figure 5). A high-efficiency ball mill, a planetary ball mill and a planetary ball mill with liquid nitrogen, produced heat, resulting in burnt specimens (Figure 5a–c). The particle size of DBM powder ground by blades was larger than 200 μm and varied in different batches (Figure 5d), while milling with diamond file made homogeneous particles of smaller size (<100 μm) (Figure 5e). Therefore, a diamond file was selected as the milling tool for further experiments.

#### 3.1.2. Characterization of DBM Powder

SEM images showed the particle size and morphology of demineralized DBM powder milled by the diamond file. The surface of the demineralized DBM was rough and the particle size of demineralized the DBMs were mostly around 26 to 75 μm (Figure 6a,b). AlCl_3_ was added to accelerate the demineralization for preparing DBM, and EDS results demonstrated the removal of calcium was complete, so that no residual calcium was observed (Figure 6c). 

Allografts and xenografts are the current alternatives for bone graft substitutes, with some side effects, such as immune responses, that can be reduced by the decellularization process. Figure 6d shows that the dsDNA content of DBM treated with different defatting and demineralization protocols did not vary and the dsDNA concentrations from all groups were below the standard criterion 50 ng/mg. Equally as important as removing immunogenic donor genetic materials is the maintenance of the osteoconductive capacity of DBM. The collagen contents from different treatment groups were above 80% of the control (nontreated DBM) as shown in Figure 6e, suggesting that certain defatting, demineralization and decellularization processes had no particular detrimental effects on collagen.

To sum up, chloroform/methanol and acidic AlCl_3_ reagents are more efficient for removal of fat, demineralizing bone specimens and maintaining collagen content. A diamond file was used following the above steps for DBM milling to prepare DBM powder. 

### 3.2. Preparation of Calcium Sulfate Hemihydrate (CaSO_4_·0.5H_2_O, CSH)

CSH is very suitable for filling and repairing bone defects and we used two different methods to prepare CSH in the α-form and β-form (Figure 2). The α-hemihydrate form was produced by the hydrothermal method with high vapor pressure of water, while the β-form was obtained from dry heat in the presence of atmospheric pressure. The α-form and β-form CSH are chemically identical but differ in their physical characteristics. The particle morphology and sizes of the α-form and β-form CSH are shown in Figure 7a. As shown, we can observe that the particle size of the α-form is from 0 µm to 15 µm, while the size of the β-form is from 16 µm to 75 µm. On the other hand, the milled and sieved DBM powder particle size is between 16 µm and 125 µm (Figure 7b). 

Preparation of the α-form CSH in a salt such as CaCl_2_ solution at atmospheric pressure was based on the dissolution-crystallization mechanism and the transformation from CSD to CSH could be completed in salt or inorganic acid solutions under normal pressure but could also create chemical residue [25,26,27]. EDS analysis was performed to examine the existence of any chloride residue in the final product, but the obtained result showed that there was no detectable chloride in the final product. The β-form CSH also contained no chloride because CaCl_2_ was not applied during production (Figure 7c). Profiles of FTIR spectra and XRD are displayed in Figure 7d,e and all of the corresponding XRD peaks were analyzed according to the database from the Joint Committee on Powder Diffraction Standards (JCPDS). The prepared CSH from this study showed typical peaks corresponding to the CSH structure, indicating complete conversion from the CSD phase to the CSH phase. 

### 3.3. DBM/CaSO_4_ Composite Bone Graft Materials

#### 3.3.1. Preparation of DBM/CaSO_4_ Composite Putty

The aim of this study was to develop a premixed bone graft putty from DBM and CSH with moldable and good handling characteristics. To this end, different ratios of DBM/CSH and various concentrations of the thickener HPMC were tested (Figure 3, Table 2 and Table 3). Glycerol (70%) was selected as a viscous carrier in these composite materials. When the ratios of DBM/CSH (either α-form or β-form) were 1:1, 7:3 or 8:2, the powder components were not mixable with a liquid component. Only the composites with DBM/CSH ratio of 9:1 could be mixed homogeneously, and this optimal ratio was used for further examination. 

When α-CSH was mixed with DBM and 70% glycerol with a liquid/powder ratio of 0.25, the appearance of the mixture was like cement and not moldable (Table 2, 0% of HPMC). To increase the viscosity of the mixture, different concentrations of HPMC as a plasticizer were added. The appearance of the final product was like clay when 2% of HPMC was used, while the addition of 4% of HPMC provided the final product with a moldable putty form (Table 2). Similar results were obtained when β-CSH was mixed with DBM and 70% glycerol with a liquid/powder ratio of 0.3, and only mixing with 4% of HPMC formed putty (Table 3). The setting time was 12 h for DBM/α-CSH putty and 4 h for DBM/β-CSH putty. 

#### 3.3.2. Properties of DBM/CaSO_4_ Composite Putty

##### Compressive Strength

The mechanical properties of DBM/α-CSH and DBM/β-CSH putty were tested by a universal mechanical testing machine. The compressive strength was 2.9 MPa for DBM/α-CSH putty and 2.73 MPa for DBM/β-CSH putty (Figure 8a), which were close to the compressive strength of the cancellous bone. 

##### Wash out Properties

The washed particle from bone graft materials after implantation could trigger inflammation and severe foreign-body responses, which could be an important issue for clinical usage [28,29]. To address this issue, a wash out test was performed and the quantitative evaluation of premixed putty before and after being soaked in PBS for 15 min is shown in Figure 8b. Wash out rate of DBM/α-CSH or DBM/β-CSH putty was estimated by calculating the weight loss of the putty. The results showed that the weight loss was 7% for DBM/α-CSH and 8.88% for DBM/β-CSH putty during the procedure, and both were lower than the reference putty (15.2% and 14.6%). This result indicates that the DBM/CSH putty developed in this study possesses good washout resistance property.

##### Biocompatibility

The cell viability of DBM/α-CSH and DBM/β-CSH composite putty extract were more than 70%, as shown in Figure 8c, which is above the ISO standard. Therefore, DBM/α-CSH and DBM/β-CSH composite putty showed low cytotoxicity and demonstrated good biocompatibility.

## 4. Discussion

In this study, porcine femur was used as the source of DBM powder for xenograft materials. Removal of fat in this procedure was achieved by incubating samples in the organic solvent chloroform/methanol, and demineralization was accomplished using an acidic AlCl_3_ reagent (Figure 1). The antigens of DBM were removed during the demineralized process and freeze-drying and sterilization reduced immunogenic response. Diethyl ester was not very effective as a defatting reagent in this study (Table 1) and was highly volatile, which made it not suitable during this procedure. As organic solvents generate offensive odors, Eagle et al. developed a cortical donor bone washing step which removes fat/lipid without the use of an organic solvent [30,31]. Bone was prepared through a series of hot water washes at 56–59 °C, centrifugation and decontamination steps, then lyophilised and ground with a compressed air milling machine. The ground bone was sieved, demineralised, freeze-dried and terminally sterilised with gamma irradiation. The resulting DBM powder produced from this procedure showed removal of DNA, extractable soluble protein, and great reduction of lipid with noncytotoxic and osteoinductive properties in an animal model [30].

There are two types of CSH, the α-form and β-form, and we successfully synthesized both types of CSH in the current study (Figure 7). α-CSH usually consists of hexagonal columnar crystalline grain, whereas the most common form, β-CSH, has a different crystallographic structure with more irregular and flaky crystals [12,13,32], consistent with the observation of CSH produced from this study (Figure 7a). The compressive strength of DBM/α-CSH putty was slightly greater than that of DBM/β-CSH putty, and DBM/α-CSH putty was also more washout-resistant than the DBM/β-CSH putty (Figure 8a). The setting time of DBM/α-CSH putty was 12 h, which was longer than for the DBM/β-CSH putty (Table 2 and Table 3). Rehydration of CSH is responsible for the setting and hardening of CS. α-CSH required much less water than β-CSH, resulting in an extremely dense dihydrate that was hard and less soluble than the β-form [13]. By contrast, the more soluble β-CSH can easily absorb moisture and thus accelerate the rate of expansion and hardening. Due to α-CSH’s better workability, and the higher mechanical strength of the hardened material, the use of α-CSH is broader than β-CSH over a wide range of fields, especially in orthopedic and other medical applications [26,33,34].

Currently there are some commercially available DBM bone substitutes, but only very few of them are combined with CS as composite materials [8]. Allomatrix^TM^ PRO-STIM^TM^ and OSTEOSET^®^ 2 DBM are manufactured by Wright Medical of the UK with calcium sulfate as a carrier. Allomatrix^TM^ consists of high DBM content (86% by volume) and uses CS as a carrier in the form of an injectable paste. It possesses great osteoconductivity and degradability and has been widely used in clinical applications such as spinal fusion, trauma surgery, or benign bone tumors [35,36,37,38]. PRO-STIM^TM^, a composite paste or putty containing 40% of DBM and CS/calcium phosphate, is usually used as bone void filler. It can accelerate healing and be osteoinductive to support bone remodeling. OSTEOSET^®^ 2 DBM is a bone graft substitute incorporating DBM (approximately 53% by volume) into OSTEOSET^®^ CS pellets which provide a combination of osteoinduction and osteoconduction for bone repair. The DBM/CSH composite putty produced in this study consisted of around 86.4% (*w/w*) DBM mixed with CSH in 70% glycerol/4% HPMC as a carrier in a moldable putty form with good biocompatibility. Further study is required to verify the osteoconductive and osteoinductive properties of the composite putty for bone healing in animal models.

## 5. Conclusions

In this study, DBM powder was obtained from porcine femur and α-/β-form CSH was successfully synthesized from CSD. We provided the optimal DBM/CSH ratio for this composite moldable putty with good biocompatibility, mechanical strength and the potential of osteoinduction/osteoconduction (Figure 9). We strongly believe that the procedure in this study might be very useful for developing and manufacturing the first commercialized DBM/CSH-based putty bone graft product domestically for clinical applications.

## Figures and Tables

**Figure 1 jfb-12-00056-f001:**
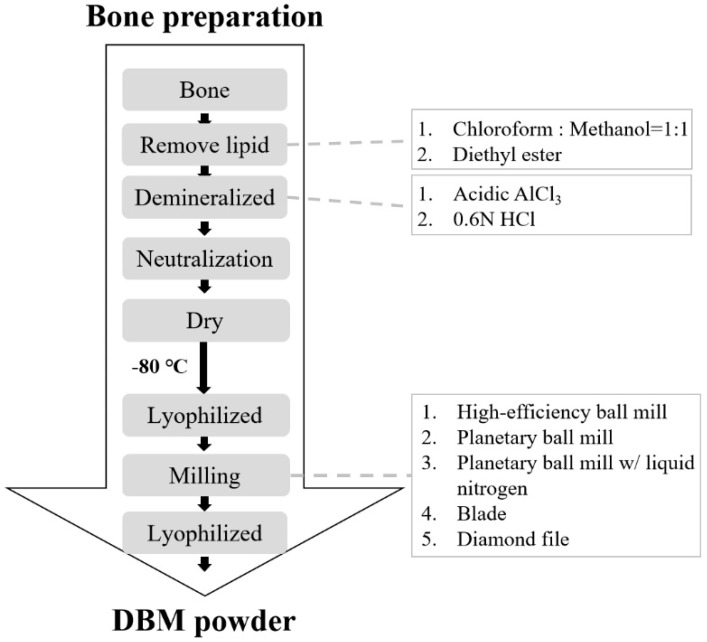
Flow chart of DBM powder preparation.

**Figure 2 jfb-12-00056-f002:**
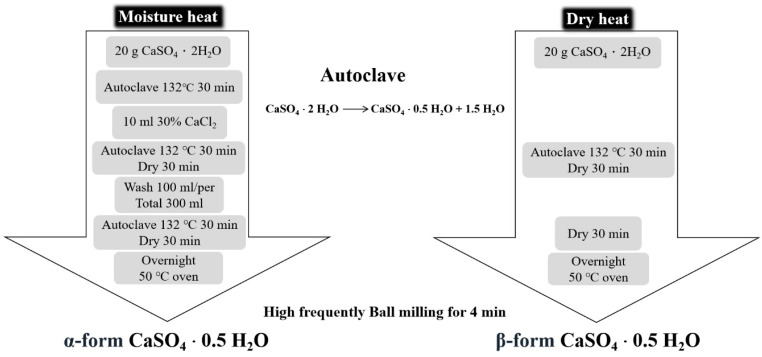
Flow chart of α-form and β-form CSH preparation.

**Figure 3 jfb-12-00056-f003:**
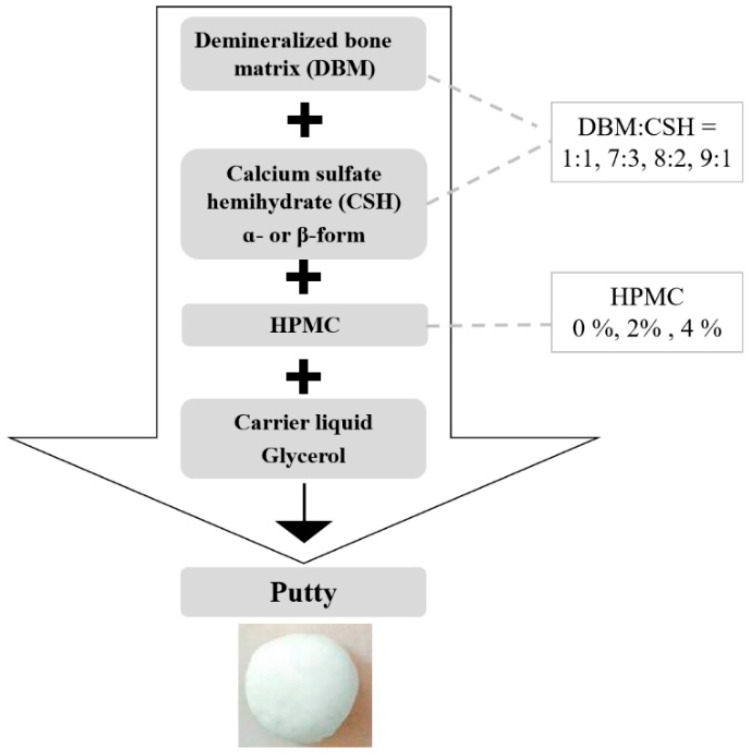
Flow chart of DBM/DSH composite putty preparation.

**Figure 4 jfb-12-00056-f004:**
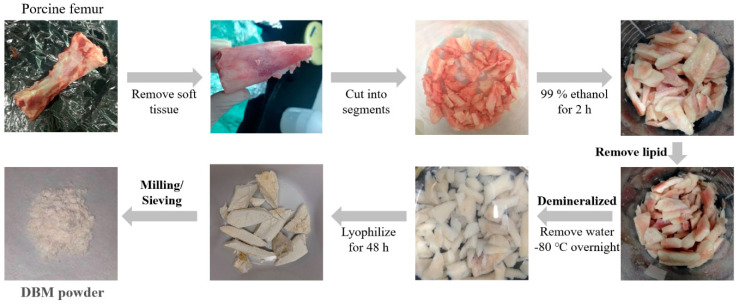
Steps for preparation of DBM powder are shown. Porcine femur was dehydrated, defatted, demineralized, freeze-dried and ground to produce DBM powder.

**Figure 5 jfb-12-00056-f005:**
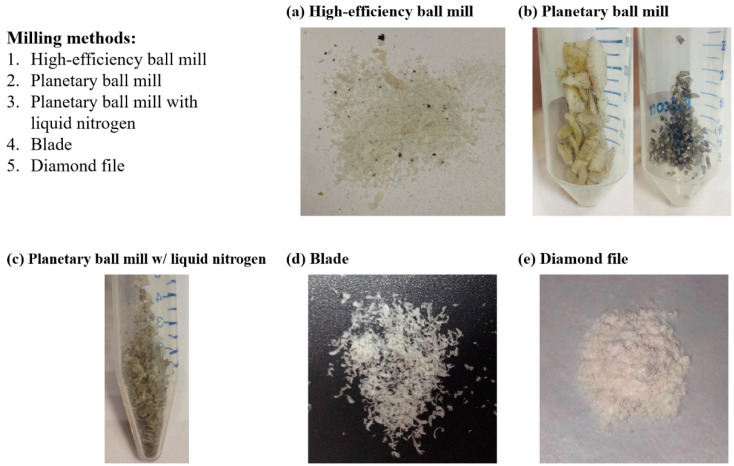
Different methods for DBM powder milling: (**a**) high-efficiency ball mill; (**b**) planetary ball mill; (**c**) planetary ball mill with liquid nitrogen; (**d**) blade; (**e**) diamond file.

**Figure 6 jfb-12-00056-f006:**
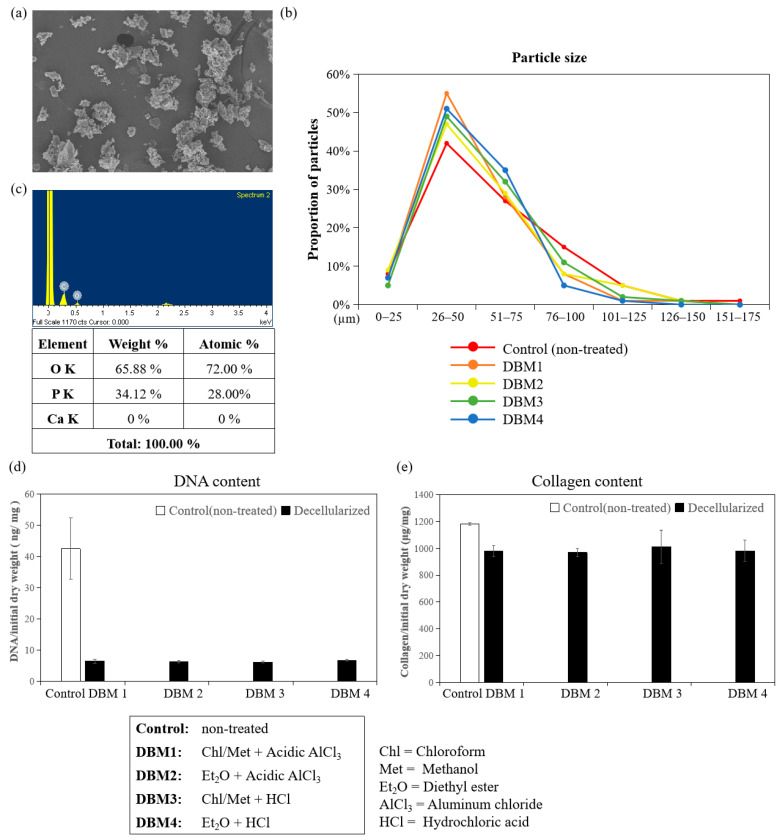
Characterization of DBM powder ground by a diamond file: (**a**) SEM images of DBM powder (DBM1 group, Table 1); (**b**) particle size of DBM powder from different treating groups; (**c**) EDS pattern of DBM; (**d**) DNA content of DBM from different treating groups; (**e**) collagen content of DBM from different treating groups.

**Figure 7 jfb-12-00056-f007:**
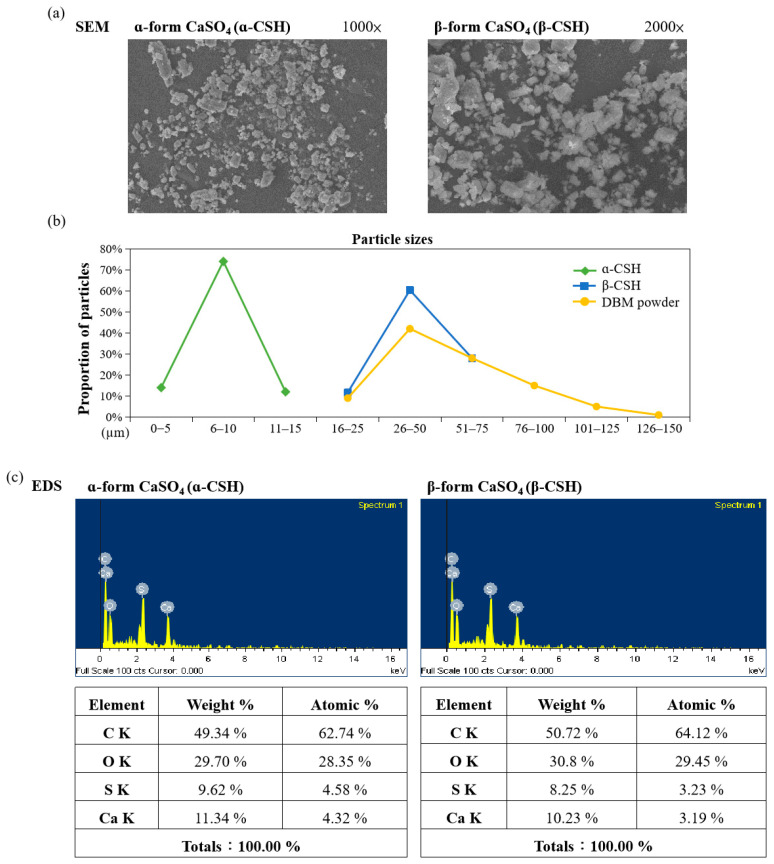
Characterization of α-CSH and β-CSH: (**a**) SEM images of CSH; (**b**) particle size of CSH and DBM; (**c**) EDS pattern of CSH; (**d**) FTIR spectra; (**e**) XRD analysis.

**Figure 8 jfb-12-00056-f008:**
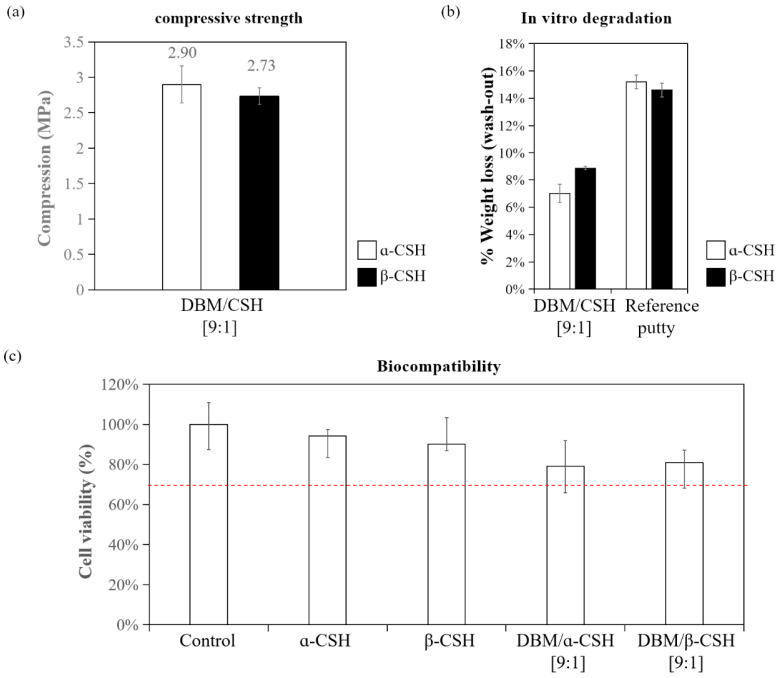
Characterization of DBM/CSH composite putty: (**a**) compressive strength; (**b**) washout resistance test; (**c**) biocompatibility. The dashed line indicates the ISO standard for cell viability (70%).

**Figure 9 jfb-12-00056-f009:**
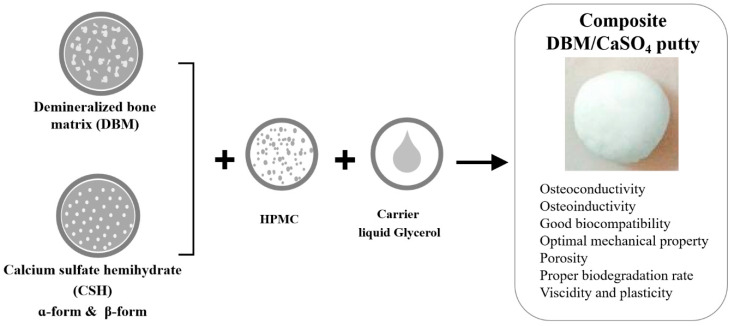
Optimal protocol provided in this study for preparing DBM/CSH composite moldable putty with good biocompatibility, mechanical strength and the potential for osteoinduction/osteoconduction.

**Table 1 jfb-12-00056-t001:** Different treating groups for production of DBM powder.

Group	Defatting Reagent	Demineralization Reagent
DBM1: Chl/Met + Acidic AlCl_3_	Chloroform/Methanol	0.5 M AlCl_3_/3% HCl/5% Formic acid
DBM2: Et_2_O + Acidic AlCl_3_	Diethyl ester	0.5 M AlCl_3_/3% HCl/5% Formic acid
DBM3: Chl/Met + HCl	Chloroform/Methanol	0.6 N HCl
DBM4: Et_2_O + HCl	Diethyl ester	0.6 N HCl

Chl = Chloroform; Met = Methanol; AlCl_3_ = Aluminum chloride; HCl = Hydrochloric acid; Et_2_O = Diethyl ester.

**Table 2 jfb-12-00056-t002:** Chemical compositions and properties of DBM/α-CSH bone graft materials.

DBM/α-CSH Composite Bone Graft Materials			
Powder	100% DBM/α-CSH [9:1] + 0% HPMC	98% DBM/α-CSH [9:1] + 2% HPMC	96% DBM/α-CSH [9:1] + 4% HPMC
Liquid	70% Glycerol + 30% water	70% Glycerol + 30% water	70% Glycerol + 30% water
L/P (Liquid/Powder)	0.25	0.25	0.25
Mixing	Yes	Yes	Yes
Self-hardening	–	–	Yes
Working time	–	–	>1 h
Setting time	–	–	12 h
Moldable	No	No	Yes
Appearance	cement	clay	Putty *

* The DBM/α-CSH putty was used for further examination.

**Table 3 jfb-12-00056-t003:** Chemical compositions and properties of DBM/β-CSH bone graft materials.

DBM/β-CSH Composite Bone Graft Materials			
Powder	100% DBM/β-CSH [9:1] + 0% HPMC	98% DBM/β-CSH [9:1] + 2% HPMC	96% DBM/β-CSH [9:1] + 4% HPMC
Liquid	70% Glycerol + 30% water	70% Glycerol + 30% water	70% Glycerol + 30% water
L/P (Liquid/Powder)	0.3	0.3	0.3
Mixing	Yes	Yes	Yes
Self-hardening	–	–	Yes
Working time	–	–	>1 h
Setting time	–	–	4 h
Moldable	No	No	Yes
Appearance	cement	clay	Putty *

* The DBM/β-CSH putty was used for further examination.

## Data Availability

The data presented in this study are available on request from the corresponding author.

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
