# Peer review of "Preparation and Characterization of Moldable Demineralized Bone Matrix/Calcium Sulfate Composite Bone Graft Materials"

_jfb, 2021, doi:10.3390/jfb12040056_

Round 1
Reviewer 1 Report
- in conclusion, authors mentioned "We provided the optimal DBM/CSH ratio for this composite moldable putty with good biocompatibility, mechanical strength and the p tentials of osteoinduction/osteoconduction ". But they have tried only one ratio 9:1. could they clarify?
- It is better to report the xrd data of milled DBM
- figure 6b, to understand better show line graph
- how did the authors prepared sample for mechanical tests. please describe in experimental section
Reviewer 2 Report
I have reviewed the mansucript Preparation and characterization of moldable demineralized 2 bone matrix/calcium sulfate composite bone graft materials. Overall, the data supports the conclusions made in the mansucript. I have only one comment -
Why reference putty (Osteo-set or Pro-stim) was not compared for the compressive strength and biocompatibility tests? What chemical compositions or interactions are most crucial for the compressive strength?
Round 2
Reviewer 1 Report
i recommend to accept the manuscript. the author answered queries and made changes in the manuscript